# Active Compounds Derived from Fuzheng Huayu Formula Protect Hepatic Parenchymal Cells from Apoptosis Based on Network Pharmacology and Transcriptomic Analysis

**DOI:** 10.3390/molecules24020338

**Published:** 2019-01-18

**Authors:** Rong Wu, Shu Dong, Fei-Fei Cai, Xiao-Le Chen, Meng-Die Yang, Ping Liu, Shi-Bing Su

**Affiliations:** 1Research Center for Traditional Chinese Medicine Complexity System, Institute of Interdisciplinary Integrative Medicine Research, Shanghai University of Traditional Chinese Medicine, Shanghai 201203, China; wurong_31@163.com (R.W.); lisadongshu@163.com (S.D.); huihuicai15@aliyun.com (F.-F.C.); 13613719846@163.com (X.-L.C.); mengdieyang@my.com (M.-D.Y.); 2E-institute of Shanghai Municipal Education Committee, Shanghai University of Traditional Chinese Medicine, Shanghai 201203, China; liuliver@vip.sina.com

**Keywords:** fuzheng huayu formula, hederagenin, luteolin, tanshinone IIA, network pharmacology, apoptosis

## Abstract

Fuzheng huayu formula (FZHY), an antifibrotic traditional Chinese medicine, is frequently used for the treatment of liver fibrosis. In this study, network analysis, transcriptomic analysis, assays of cell apoptosis, viability and protein expression were used for investigating the effects and mechanisms of compounds derived from FZHY on hepatic parenchymal cell (HPC) protection and hepatic stellate cell activation. Network pharmacology analysis found that 6 major compounds and 39 potential targets were important network nodes. Our analysis predicted that the active compounds of FZHY, including hederagenin, luteolin and tanshinone IIA inhibited cell apoptosis (*p* < 0.05), increased PI3K expression and reduced cleaved caspase 3 expression and the Bax/Bcl-w ratio (*p* < 0.05) in L02 cells that had apoptosis induced by TNF-α. Few significant changes caused by FZHY, hederagenin, luteolin and tanshinone IIA were observed in hepatic stellate Lx2 cells upon TGF-β1 induction. These data suggest that FZHY is active against liver fibrosis, protects hepatic parenchymal cells from apoptosis, and recovers liver function, possibly through the effects of its active compounds hederagenin, luteolin and tanshinone IIA and is involved in the inhibition of apoptosis in HPCs, possibly through regulating the PI3K, ERK, cleaved caspase 3 and Bax/Bcl-w levels.

## 1. Introduction

Liver fibrosis, characterized by fibrosis and the accumulation of collagen [1,2], is a consequence of liver damage from various sources, including metabolic disorders, hepatic parenchymal cell (HPC) damage, autoimmune diseases, alcohol abuse and hepatic viral infection [3]. As chronic liver disease worsens, further hepatocellular dysfunction, chronic inflammation and progressive fibrosis can turn liver fibrosis into liver cirrhosis. Liver cirrhosis can induce complications such as ascites, bleeding from dilated veins in the esophagus, hepatic encephalopathy and liver cancer [4]. Thus, anti-liver fibrosis capability is vital to the study of cirrhosis treatment. In recent years, the clinical treatment for liver fibrosis has mainly addressed hepatic stellate cell (HSC) activation, apoptosis protection in HPCs, inflammation and the regulation of hepatic extracellular matrix (ECM) metabolism; however, there has not been a significant increase in novel drugs for this condition [4,5].

As the foundation of liver function, HPCs make up 70–85% of the liver’s mass. They are involved in protein synthesis, protein storage, transformation of carbohydrates, etc. Therefore, damage of HPCs is the necessary prerequisite of liver fibrosis. HPCs apoptosis plays a role in the process of liver fibrosis. Apoptosis caused by apoptosis-inducing factor and inflammatory factors can induce the damage and abscission of HPCs. Tumor necrosis factor alpha (TNF-α), a chemokine, takes part in most apoptosis regulation. The increase of TNF-α by damage or inflammation in liver can induce HPCs apoptosis.

HSCs are the major cell type involved in liver fibrosis, and they are in a quiescent state in normal liver. When the HPCs are damaged, they change into an activated state [4,5]. The activation of HSCs then leads to the repair of fibrotic tissue, secreting collagen scar tissue and the proliferation of scar tissue. Thus, the activation of HSCs is also the necessary prerequisite of liver fibrosis. Transforming growth factor beta 1 (TGF-β1), performs cell proliferation and differentiation. As a reaction to liver injury, the increase of TGF-β1 can cause activation of HSCs [6]. Because of the complex mechanism of liver fibrosis, few effective drugs with the therapeutic strategy of repairing liver damage or reversing progressive fibrosis have been reported [6]. Fortunately, with the gradual clarification of the underlying mechanisms of this disease, some treatment strategies and drugs from traditional Chinese medicine (TCM) have been reported [4,7].

Fuzheng huayu formula (FZHY), an SFDA-approved antifibrotic medicine, consists of six Chinese herbs. It contains *Salvia miltiorrhiza* Bunge, fermentation mycelium powder [*Cordyceps sinensis* (Berk.) Sae.], *Schisandra chinensis* (Turcz.) Baill., pollen pini (*Pinus massoniana* Lamb.), semen persicae [*Prunus persica* (L.) Batsch] and *Gynostemma pentaphyllum* (Thunb.) Makino. In China, FZHY is frequently used for the treatment of liver fibrosis. Recent pharmacological studies have reported that it is effective in reducing hyaluronidase, alleviating liver fibrosis [8,9] and relieving complications [10,11]. Some studies have reported that the mechanisms of FZHY are probably associated with many signaling pathways [12,13,14]. In our previous study, FZHY administration significantly improved liver function, and alleviated hepatic inflammatory and fibrotic changes in the livers of CCl_4_-treated rats [4]. Its anti-fibrotic effect was associated with retinol metabolism, xenobiotic metabolism by cytochrome P450, et al. However, the molecular mechanisms and active compounds of FZHY remain unclear.

The development of new fields in modern pharmacological studies, such as network pharmacology, transcriptomics, and systems biology, has provided novel approaches for TCM research [4]. These techniques can be used for clarifying the molecular mechanisms, including multiple targets and multiple effects, of many compounds from a system perspective [15]. The evaluation of high-throughput biological data and network pharmacology has been proven to be a suitable method for exploring synergistic effects in complex diseases and in the mechanisms of TCMs [16,17]. It also provides an approach for the further investigation of FZHY to alleviate liver fibrosis and recover liver function.

Therefore, we performed this study to evaluate the anti-liver fibrosis and HPC-protecting effects of FZHY and to elucidate the molecular mechanisms of FZHY, hederagenin, luteolin and tanshinone IIA on HPC protection through the use of transcriptomic data combined with network pharmacology-based analysis and pharmacological experimental verification.

## 2. Results

### 2.1. Differentially Expressed Genes (DEGs), Direct Related Genes of DEGs, and Potential Targets Analysis

In a previous study, the liver tissues of the control rats, model rats (CCl_4_-treated) and FZHY rats (CCl_4_ and FZHY-treated) were collected [4]. The liver tissues of model rats (CCl_4_-treated) and FZHY rats were used for mRNA microarray assays. In this study, a total of 117 DEGs were collected by mRNA microarrays data analysis (*n* = 3). A heatmap and the most regulated DEGs are shown in Figure 1a,b. According to human protein–protein interactions (PPIs), 262 directly related genes of DEGs were obtained for finding the potential targets. Chemical compounds and targets of FZHY were collected. A Venn analysis of the genes directly related to DEGs and targets of FZHY was performed to obtain potential targets of FZHY that may contribute to an anti-fibrotic effect. A total of 62 potential targets were found, as shown in Figure 1c. KEGG pathway analyses were carried out to evaluate the mechanisms of the 62 potential targets of FZHY. The main pathways of the potential targets are shown in Figure 1d. The 62 targets were enriched in 180 pathways. They participate in many bioprocesses, including apoptosis, ECM organization, response to oxidative stress, inflammatory response and drug metabolism. All DEGs, targets of FZHY, genes directly related to DEGs, potential targets of FZHY with anti-fibrotic effects, pathways of potential targets and dataset of raw microarray data are listed in Appendix A, respectively.

### 2.2. Network Pharmacology Analysis

After evaluating the topological parameters in a basic network, an herb-compound-target-signaling pathways network was constructed (Figure 2a). This network is composed of 5 herbs, 6 compounds, 39 targets and 12 pathways. The data of this network are listed in Appendix A. Through a comprehensive evaluation for the topological parameters and stabilization of an herb-compound-target-signaling pathway, putative major signaling pathways of FZHY were constructed, as shown in Figure 2b.

### 2.3. Effects of FZHY-Medicated Serum and Its Major Compounds on L02 and Lx2 Cell Viability

The effect of FZHY-medicated serum and its major compounds on the viability of L02 and Lx2 cells was evaluated (Figure 3). It is difficult to carry out an in vitro experiment using unsaturated fatty acids; therefore, we did not validate arachidonic acid. In the medicated serum, TNF-α + actinomycin D (ACTD) significantly inhibited the viability of L02 cells at 24 h, 48 h and 72 h (*p* < 0.01; Figure 3a); 10% FZHY-medicated serum increased the viability of L02 cells at 24 h (*p* < 0.05; Figure 3a). TGF-β1 significantly increased the viability of Lx2 cells at 24 h, 48 h and 72 h (*p* < 0.01; Figure 3b); 10% FZHY-medicated serum reduced the viability of Lx2 cells at 24 h, 48 h and 72 h (*p* < 0.01; Figure 3b). Only 10% FZHY-medicated serum reduced the viability of Lx2 cells at 48 h and 72 h (*p* < 0.05; Figure 3b). Regarding the major compounds, beta-sitosterol inhibited the viability of L02 cells at 100 µmol/L; kaempferol inhibited the viability of L02 cells at 50 µmol/L; hederagenin and tanshinone IIA inhibited the viability of L02 cells at 25 mol/L; and luteolin inhibited the viability of L02 cells at 6.25 µmol/L (*p* < 0.05; Figure 3c). Kaempferol, tanshinone IIA and beta-sitosterol inhibited the viability of Lx2 cells at 50 µmol/L; hederagenin inhibited the viability of Lx2 cells at 25 µmol/L; and luteolin inhibited the viability of Lx2 cells at 12.5 µmol/L (*p* < 0.05; Figure 3d). In this work, the concentration of individual compounds is not equal to their concentration in FZHY extracts or FZHY-medicated serum. The concentration of individual compounds in FZHY-medicated serum may be not enough to inhibit viability in L02 cells. Therefore, only individual compounds inhibit viability in L02 cells. The IC50 values of the major compounds with L02 and Lx2 cell lines are shown in Table 1.

### 2.4. Effects of FZHY-Medicated Serum and Its Major Compounds on L02 and Lx2 Cell Apoptosis

The effect of FZHY-medicated serum and its major compounds on the apoptosis of L02 and Lx2 cells was assayed, and the results of the Muse^TM^ Annexin V Dead Cell Kit are displayed by histograms, as shown in Figure 4a–c. In the medicated serum, compared to the control group, TNF-α + ACTD significantly induced apoptosis in L02 cells (*p* < 0.01; Figure 4a,b), and 10% FZHY-medicated serum reduced the total apoptotic rates (*p* < 0.05; Figure 4a). Regarding the major compounds, TNF-α + ACTD significantly increased the total apoptotic rates in L02 cells (*p* < 0.01; Figure 4c), and the total apoptotic rates in the tanshinone IIA, hederagenin and luteolin groups were reduced (*p* < 0.01; Figure 4c). Because the FZHY-medicated serum showed no effect on the apoptosis of Lx2 cells, we did not carry out an evaluation of the major compounds on Lx2 cell apoptosis.

### 2.5. Effects of FZHY-Medicated Serum and Its Major Compounds on Potential Targets

An evaluation of the potential targets that were predicted to validate and reveal the mechanism of FZHY is shown in Figure 5. TNF-α + ACTD dramatically reduced the expression of PI3K and *p*-ERK1/2, increased cleaved caspase 3 and Bax/Bcl-w levels (*p* < 0.05; Figure 5a,b); 10% FZHY-medicated serum, tanshinone IIA, hederagenin and luteolin reduced cleaved caspase 3 and Bax/Bcl-w levels, and 10% FZHY-medicated serum, hederagenin and luteolin increased the expression of PI3K (*p* < 0.05; Figure 5a,b). In the Lx2 cells, TGF-β1 significantly decreased the *p*-ERK1/2 levels (*p* < 0.01; Figure 5c,d). Compared with the expression levels in the TGF-β1-treated group, 10% FZHY-medicated serum reduced the expression of PI3K, collagen I and αSMA; tanshinone IIA, hederagenin and luteolin decreased the expression of *p*-ERK1/2; hederagenin and luteolin reduced the collagen I level, and hederagenin decreased the levels of αSMA (*p* < 0.01; Figure 5c–f).

## 3. Discussion

Liver fibrosis resulting from chronic liver injury is a major causes of morbidity and mortality worldwide [18]. In clinical treatment, liver fibrosis therapies are focused on reducing the possibility of liver transplantation and improving the quality of life [19]. The aim of most recent studies is to reverse fibrosis and prevent liver damage [8]. As yet, few effective treatments have been reported. Advanced liver cirrhosis is still incurable and can lead to liver failure and death [20]. Although the molecular mechanisms of FZHY in HPC protection and its antifibrotic effects are not fully clear, many reports have shown its antifibrotic effects without adverse effects [9,21,22]. Thus, evaluating the antifibrotic and HPC-protecting effects of FZHY and clarifying its mechanisms of action is important. Our previous study [4] revealed that FZHY could reduce the injury and abscission of normal hepatocytes, and relieve fibrosis, oedema and inflammation [23]. 

Transcriptomic technologies can measure the expression of genes, giving information on how genes are regulated in tissues [24]. It has been instrumental in the understanding of drug effects as a whole. Network pharmacology, as a novel research method for Chinese herbal formulas, can help to identify putative targets, active compounds and pharmacological mechanisms in TCM research [25,26]. In studies of complex diseases, transcriptomics has been widely applied with network pharmacology analysis to clarify molecular mechanisms. In this work, we overlapped the targets of FZHY and genes directly related to DEGs of FZHY-treated rats, resulting in 62 potential targets of the molecular mechanism of the antifibrotic effect by FZHY. At the same time, we carried out an enrichment analysis to find potential pathways of the FZHY antifibrotic effects. Major compounds and targets were selected using topological parameters and a structure herb-compound-target-signaling pathway network of FZHY, which resulted in the selection of 6 compounds, 39 potential targets and 12 pathways. Previous studies have reported that the selected major compounds, such as kaempferol and tanshinone IIA, inhibit the proliferation of HSCs [21], luteolin has hepatoprotective activity through the regulation of SREBP-1c phosphorylation and AMP-activated protein kinase [27], beta-sitosterol prevents hepatotoxicity [28], and hederagenin can attenuate non-alcoholic hepatic steatosis [29].

After the evaluation of the primary network nodes and shared signaling pathways of potential targets of FZHY and liver fibrosis, we found that the affected signaling pathways of liver fibrosis treated by FZHY mainly include intracellular signaling in apoptosis, such as the PI3K-Akt, p53, TNF and MAPK-ERK pathways. These pathways [30] are also associated with liver fibrosis. However, distinguishing inhibitory effects from activation effects is difficult in network pharmacology. Moreover, this type of analysis is susceptible to influence by prediction tools. Thus, it is essential to validate the predictions.

Both HPC and HSC viabilities are closely related to their proliferation and the inhibition of apoptosis. Thus, the cell viability assay is commonly used to evaluate the efficacy of antifibrotic and hepatoprotective drugs. In studies of liver diseases, Lx2 cells (a type of HSC) and L02 cells (a type of HPC) are frequently used in vitro. In this study, L02 cells (treated with TNF-α) and Lx2 cells (treated with TGF-β1) were used for simulating the HPCs apoptosis and HSCs activation in the process of liver fibrosis, respectively. 10% FZHY-medicated serum increased the viability of L02 cells treated with TNF-α and reduced the viability of Lx2 cells treated with TGF-β1. These results show that FZHY can inhibit the proliferation of HSCs induced by TGF-β1 and attenuate the decrease of HPG viability caused by TNF-α. With the major compounds, significant changes at low concentrations in both Lx2 and L02 cell viabilities were found for the hederagenin, luteolin and tanshinone IIA groups. Therefore, we chose these 3 compounds for use in the follow-up experiments. For HPCs, we used low concentrations of the 3 compounds and did not inhibit L02 cell proliferation. In addition, according to the network pharmacology prediction, FZHY may be involved in the protection or induction of apoptosis processes.

Apoptosis, a programmed cell death process, is an important biological process in liver fibrosis [31]. In liver fibrosis therapies, studies have been focused on the suppression of apoptosis in HPCs and the induction of apoptosis in HSCs. In this work, the results of the apoptosis assays showed that 10% FZHY-medicated serum can reduce the total apoptosis rate induced by TNF-α in L02 cells. Since no change was observed in Lx2 cells treated with 10% FZHY-medicated serum, we only used compounds in the L02 apoptosis assay. Hederagenin, luteolin and tanshinone IIA markedly reduced the total apoptosis rate of L02 cells induced by TNF-α. These data indicated that FZHY, hederagenin, luteolin and tanshinone IIA can protect the HPCs from apoptosis, leading to liver damage recovery, which was in line with our prediction. However, these results showed that FZHY may not affect the apoptosis process in HSCs. We surmised that FZHY reduces HPC apoptosis, which leads to recovering liver function and relieving liver damage, with an anti-liver fibrosis effect. At the same time, in our prediction, FZHY primarily regulates the PI3K-Akt, TNF and MAPK-ERK pathways to suppress HPC apoptosis. We determined the protein expression of potential targets in these apoptosis-related pathways, which were obtained from the network pharmacology analysis, to validate the prediction.

The PI3K-Akt pathway is a classic pathway in the progression of liver fibrosis and is involved in the apoptosis and proliferation of HSCs and HPCs. The low-expression of the PI3K-Akt pathway can induce apoptosis [32], which is related to HPC damage. PI3K activates Akt to affect BAX, NF-κB and mTOR, which leads to indirectly or directly promoting the proliferation and inhibition of apoptosis [33]. Therefore, upregulating the low-expression of PI3K may restore the balance of apoptosis and proliferation in HPCs, leading to liver recovery. In liver fibrosis, the low-expression of the MAPK-ERK pathway can promote cell cycle blockage, induce apoptosis, reduce epithelial-mesenchymal transition (EMT) and inhibit proliferation. Phospho-ERK1/2 can activate related genes, including the Bcl-2 protein family, to promote anti-apoptosis. Thus, evaluation of phospho-ERK1/2 expression can assess apoptosis and cell cycle blockage in HPCs.

The TNF pathway, containing caspases, pro-apoptotic and anti-apoptotic members, is involved in liver fibrosis. As the classic marker of apoptosis, caspase 3 is activated by mitochondrial pathways and death ligands. Because of its activation by cleaving, assessing cleaved caspase 3 can determine apoptosis. Bax promotes the progression of apoptosis by forming a heterodimer with an apoptotic member. In contrast, Bcl-w, a member of the Bcl-2 protein family, can suppress the activation of caspases, leading to an anti-apoptosis effect. Thus, the Bax/Bcl-w ratio is a suitable parameter to evaluate the anti-apoptosis effect. In this work, alterations in PI3K, *p*-ERK1/2, ERK1/2, cleaved-caspase 3, caspase 3, and the Bax/Bcl-w ratio after treatment with FZHY and its 3 major compounds were observed. The results of the apoptosis assays and western blot data of Lx2 with TGF-β1 treatment showed that FZHY and its compounds such as hederagenin (derived from semen persicae), luteolin and tanshinone IIA (both derived from *Salvia miltiorrhiza* Bunge) inhibit HSC activation. The L02 data showed that FZHY, hederagenin, luteolin and tanshinone IIA protect HPCs from apoptosis by regulating the apoptosis-related pathways, leading to a recovery of liver function, lessen HPC damage and improve the quality of life of patients with liver fibrosis. The comprehensive effects of hederagenin, luteolin and tanshinone IIA are consistent with those of FZHY treatment, as well as the results of the network pharmacology prediction.

In conclusion, this study shows that FZHY can prevent HPC apoptosis, which is connected to its modulatory effects on PI3K, ERK, cleaved caspase 3 and Bax/Bcl-w. According to the network pharmacology prediction and experimental verification in vitro, these effects may be derived from hederagenin, luteolin and tanshinone IIA. This study showed that FZHY and its compounds, such as hederagenin, luteolin and tanshinone IIA, might be able to be used in early liver fibrosis due to their HPC protective effects. These results may contribute to the development of clinical applications of FZHY. However, further research is required to investigate the micro-synergistic effects of multiple compounds and metabolites using their concentration in FZHY extracts or FZHY-medicated serum, both in vitro and vivo.

## 4. Materials and Methods

### 4.1. Reagents

Kaempferol (HPLC ≥ 98%), hederagenin (HPLC ≥ 98%), beta-sitosterol (HPLC ≥ 98%), luteolin (HPLC ≥ 98%), tanshinone IIA (HPLC ≥ 98%), tanshinol B (HPLC ≥ 98%), protocatechualdehyde (HPLC ≥ 98%) and salvianolic acid B (HPLC ≥ 98%) were obtained from the Standardization Research Center of TCM (Shanghai, China). The concentration of DMSO used in this study was <0.1%. The compounds were stable under the conditions used in the study. All chemical structures are shown in Appendix A.

Tetrazolium compound MTS was obtained from Promega (Madison, WI, USA). Foetal bovine serum (FBS) and Dulbecco’s modified Eagle’s medium (DMEM) were obtained from Gibco Life Technologies (Waltham, MA, USA). TNF-α and ACTD were obtained from Peprotech (Rocky Hill, NJ, USA). Muse^TM^ Annexin V Dead Cell kits were obtained from EMD Millipore (Burlington, MA, USA). Antibodies against GAPDH, cleaved caspase-3, caspase-3, Bax, Bcl-xL, PI3K, αSMA, collagen I, *p*-ERK1/2 and ERK1/2 were obtained from Cell Signaling Technology (Danvers, MA, USA).

### 4.2. FZHY Preparation and FZHY-Medicated Serum Preparation

FZHY formula was obtained from Shanghai HuangHai Pharmaceutical Co., Ltd. (Shanghai, China). It contained 8 g of *Salvia miltiorrhiza* Bunge, 4 g of fermentation mycelium powder [*Cordyceps sinensis* (Berk.) Sae.], 2 g of *Schisandra chinensis* (Turcz.) Baill., 2 g of pollen pini (*Pinus massoniana* Lamb.), 2 g of semen persicae [*Prunus persica* (L.) Batsch] and 6 g of *Gynostemma pentaphyllum* (Thunb.). All herbs were fully validated using mpns.kew.org. The FZHY formula was concentrated to 0.35 g crude drug/mL.

Male Sprague-Dawley (SD) rats [4,34] in the FZHY (*n* = 8) and normal control (*n* = 8) groups were assigned according to random digit. Rats in the FZHY group received intragastric administration of FZHY (3.5 g/kg/day) for 3 days, while the control group received the same volume of saline for 3 days. After the last administration, the rats were anaesthetized. The blood from the abdominal aorta was centrifuged into serum and preserved at −80 ℃. The FZHY-medicated serum was prepared according to Meng et al. [34].

This study was carried out in accordance with the recommendations of the Care and Use of Laboratory Animals published by the U.S. National Institutes of Health (NIH Publication No. 85-23, revised 1996) and the Animal Care and Use Committee of the Shanghai University of Traditional Chinese Medicine. The protocol was approved by the Animal Care and Use Committee of the Shanghai University of Traditional Chinese Medicine (the project identification code: SZY201709014, date of approval: 10 September 2017).

HPLC-MS MRM chromatograms of FZHY extracts and FZHY-medicated serum samples are shown in Appendix A. Tanshinol B, protocatechualdehyde and salvianolic acid B were used as the quality control indicators of FZHY extracts and FZHY-medicated serum samples.

### 4.3. Animal Administration and mRNA Microarrays

In a previous study, male Wistar rats (8 weeks, 170–190 g) [4,34] were assigned to two groups: the control group (*n* = 10) and the CCl_4_-treated group (*n* = 20). Rats in the CCl_4_-treated group were injected with 50% CCl_4_ in an olive oil solution (i.p.) at a dose of 1 mL/kg every 3 days for nine weeks, and rats in the control group received the same volume of olive oil (the solvent for CCl_4_). At the sixth week, the CCl_4_-treated rats were allocated to two groups (according to random digit.): the CCl_4_-treated group and the FZHY group (*n* = 10). Rats in the FZHY group were gavaged using FZHY (0.35 g/kg/day) for 4 weeks; the other rats received the same volume of saline [4]. At end of the experiment, rats were sacrificed with 20% urethane and liver tissues were frozen in liquid nitrogen and preserved at −80 °C for proteome analysis and mRNA microarray assays.

Total RNA in liver tissues from the model group and FZHY-treated group (*n* = 3) were isolated by TRIzol reagent, and quality control was conducted by an Agilent Bioanalyser 2100 (Agilent Technologies, Santa Clara, CA, USA). The slides were scanned by an Agilent microarray scanner (Agilent Technologies). The raw data were evaluated by Feature Extraction software 10.7 (Agilent Technologies and normalized by Quantile algorithm, Gene Spring Software 11.0 (Agilent Technologies).

Transcriptomics data was obtained from our previous study [4] and analyzed with the OmicsBean data analysis tool (2.0, www.omicsbean.cn/). The DEG criteria included the following: fold change > 1.5 and *p*-value < 0.05. The directly related genes of DEGs were identified and collected.

### 4.4. Collection of Formulation Compounds and Targets

The chemical compounds and the compound-related targets were collected using the Traditional Chinese Medicine (TCM) database of Taiwan (http://tcm.cmu.edu.tw/) and the Traditional Chinese Medicine Systems Pharmacology (TCMSP) database (http://lsp.nwu.edu.cn/tcmsp.php). The human protein-protein interactions (PPIs) were obtained from the STRING (https://string-db.org/), DisgeneT (http://www.disgenet.org), OmicsBean (www.omicsbean.cn/) and Reactome (www.reactome.org/) databases. The compounds were screened for both the pharmacokinetic and pharmacodynamic properties (oral bioavailability (OB) > 30%; drug-likeness (DL) > 0.18). An integrative analysis of DEGs, the directly related genes and FZHY targets was performed using a Venn analysis. The overlap was presumed to be the group of potential targets, and these were used in the network pharmacology analysis. All pathways were obtained by OmicsBean (www.omicsbean.cn/).

### 4.5. Network Pharmacology Analysis

The compound-target networks of FZHY were constructed by Cytoscape software (Version 3.2.0, National Institute of General Medical Sciences, Bethesda, MD, USA). Related parameters were calculated to explore significant nodes [35]. An evaluation of the network stabilization was carried out according to Liu Y et al. [36].

### 4.6. MTS Assay

L02 and Lx2 cells were purchased from the Type Culture Collection of the Chinese Academy of Sciences (Shanghai, China) and were cultured as previously described [37]. L02 and Lx2 cells were cultured at 37 °C in a 96-well plate (5.0 × 10^3^ cells/well) for 12 h, followed by incubation with kaempferol, hederagenin, beta-sitosterol, tanshinone IIA (6.25–200 µmol/L) and luteolin (3.125–200 µmol/L) for 24 h, or with 10% FZHY-medicated serum for 24 h, 48 h and 72 h to observe cell viability; 20 μL of MTS was then added. After 3.5 h, the absorbance at 490 nm was measured. All assays were repeated at least 3 times. The passages of L02 cells is 4th generation; the passages of Lx2 cells is 7th generation.

### 4.7. Apoptosis Assay

L02 and Lx2 cells were cultured in a 6-well plate (3 × 10^5^ cells/well) for 12 h, followed by incubation with hederagenin (12.5 µmol/L), luteolin (3.125 µmol/L), tanshinone IIA (12.5 µmol/L), TNF-α (20 ng/mL) + ACTD (200 ng/mL) or 10% FZHY-medicated serum for 24 h in L02 cells, and hederagenin (25 µmol/L), luteolin (12.5 µmol/L), tanshinone IIA (25 µmol/L), TGF-β1 (20 ng/mL) or 10% FZHY-medicated serum for 24 h with TGF-β1 (20 ng/mL) in Lx2 cells. The concentration of TNF-α and ACTD was carried out according to Tao YY et al. [38]. Cell apoptosis was then detected using a Muse^TM^ Annexin V Dead Cell kit (Merck-Millipore, Burlington, MA, USA). The detection assays were repeated 3 times.

### 4.8. Western Blot Assay

L02 and Lx2 cells were cultured in a 6-well plate at a density of 1.0 × 10^6^ cells/well for 12 h, followed by incubation with hederagenin (12.5 µmol/L), luteolin (3.125 µmol/L), tanshinone IIA (12.5 µmol/L), TNF-α (20 ng/mL) + ACTD (200 ng/mL) or 10% FZHY-medicated serum for 24 h in L02 cells, and hederagenin (25 µmol/L), luteolin (12.5 µmol/L), tanshinone IIA (25 µmol/L), TGF-β1 (20 ng/mL) or 10% FZHY-medicated serum for 24 h with TGF-β1 (20 ng/mL) in Lx2 cells. The cells were then harvested. The analyses were carried out according to previous literature [36], and the assays were repeated at least 3 times.

### 4.9. Statistical Analysis

One-way analysis of variance (ANOVA) and rank-sum tests were evaluated (SPSS 18.0 software, IBM, Armonk, NY, USA). A *p*-value of *p* < 0.05 was considered to be statistically significant.

## Figures and Tables

**Figure 1 molecules-24-00338-f001:**
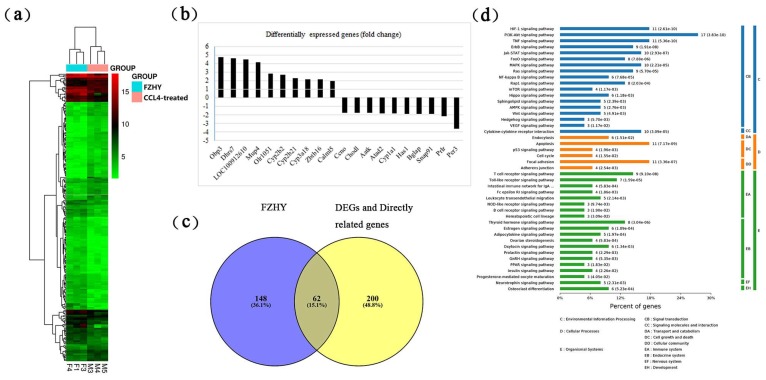
DEGs and potential target analysis. (**a**) Heatmap of DEGs. (**b**) Topmost regulated DEGs. (**c**) 62 potential targets from Venn analysis. (**d**) The main pathways of potential targets. FZHY, FZHY formula. M, model rats (CCl_4_-treated); F, FZHY rats (CCl_4_ and FZHY-treated). Percentages in the Venn diagram: related genes/total genes. The numbers at the end of the bars represent for the *p*-value of pathways enrichment analysis.

**Figure 2 molecules-24-00338-f002:**
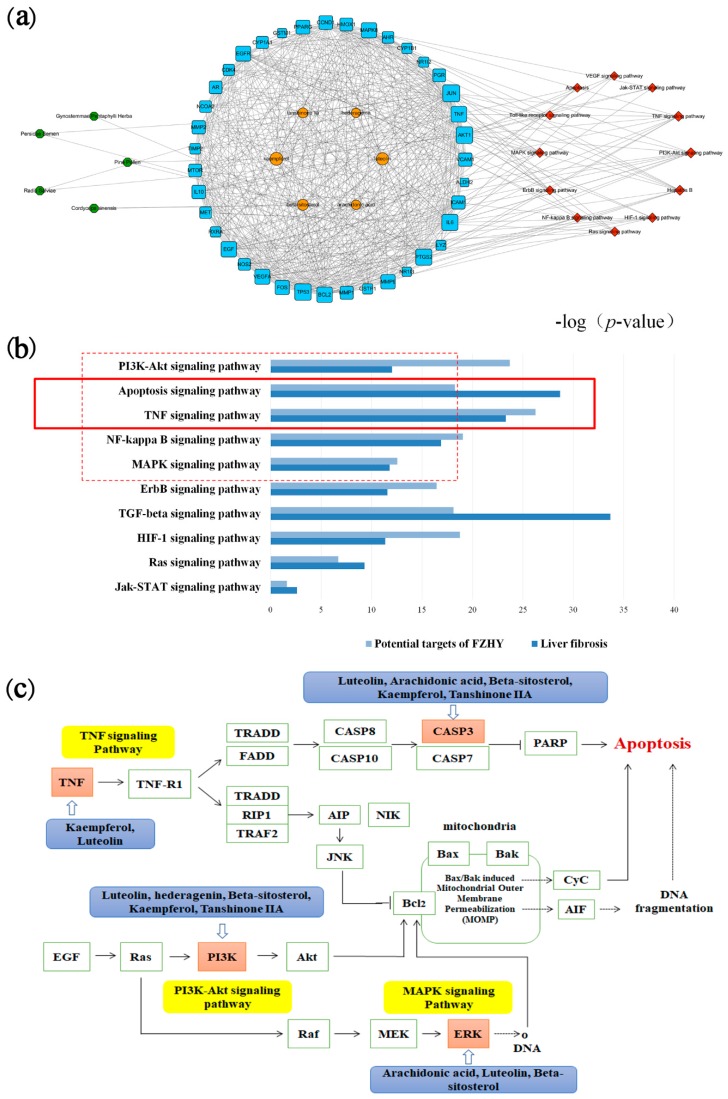
Network pharmacology analysis. (**a**) Herb-compound-target-signaling pathway of FZHY. The green octagon nodes represent herbs, the orange ellipse nodes represent compounds, the blue square nodes represent targets and the red diamond nodes represent signaling pathways. (**b**) Top 10 shared signaling pathways of potential targets of FZHY and liver fibrosis. (**c**) Putative major signaling pathways of FZHY. The blue rectangles represent compounds, the yellow rectangles represent signaling pathways, the squares represent targets, and the orange squares represent predicted targets. FZHY, FZHY formula.

**Figure 3 molecules-24-00338-f003:**
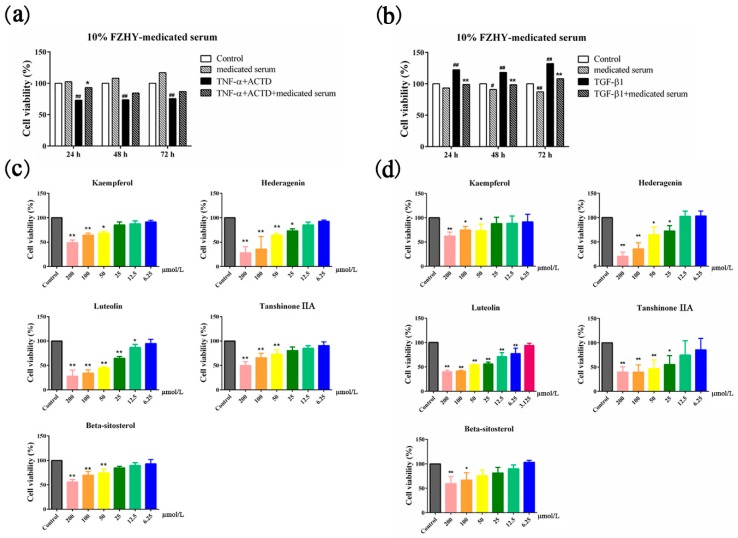
(**a**) Effects of FZHY-medicated serum on L02 cells viability at 24, 48, and 72 h. (**b**) Effects of FZHY-medicated serum on Lx2 cells viability at 24, 48, and 72 h. ^#^
*p* < 0.05, ^##^
*p* < 0.01, compared with the control group. * *p* < 0.05, ** *p* < 0.01, compared with the TNF-α + ACTD or TGF-β1 group. (**c**) Effects of kaempferol, hederagenin, luteolin, tanshinone IIA and beta-sitosterol on L02 cells viability at 24 h. (**d**) Effects of kaempferol, hederagenin, luteolin, tanshinone IIA and beta-sitosterol on Lx2 cell viability at 24 h. FZHY, FZHY formula. * *p* < 0.05, ** *p* < 0.01, compared with the control group. Repeated 4 times.

**Figure 4 molecules-24-00338-f004:**
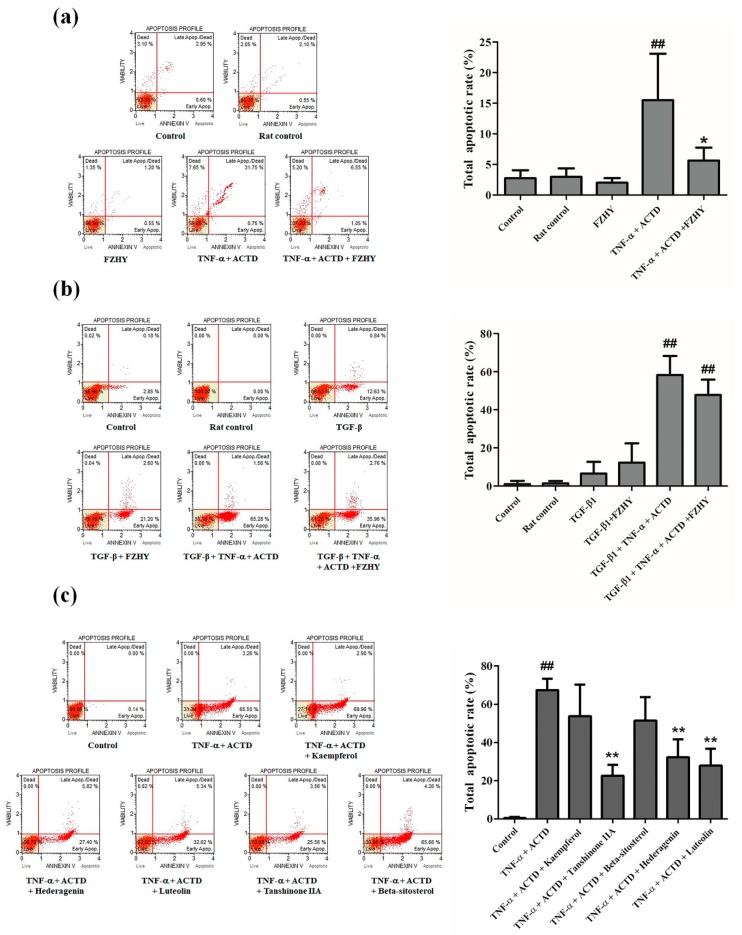
Apoptosis assay. (**a**) Effects of FZHY-medicated serum on L02 cells apoptosis. (**b**) Effects of FZHY-medicated serum on Lx2 cells apoptosis. (**c**) Effects of kaempferol, hederagenin, luteolin, tanshinone IIA and beta-sitosterol on L02 cells apoptosis. FZHY, FZHY formula. ^##^
*p* < 0.01, compared with the control group. * *p* < 0.05, ** *p* < 0.01, compared with the TNF-α + ACTD group. Repeated 4 times.

**Figure 5 molecules-24-00338-f005:**
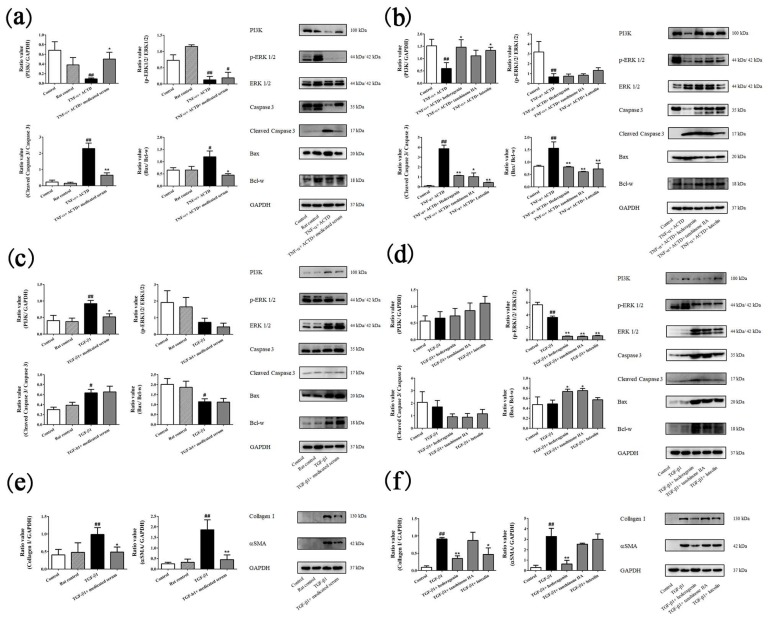
Validation of potential targets with western blot. (**a**) Effects of FZHY-medicated serum on the expression of potential targets in L02 cells. (**b**) Effects of tanshinone IIA, hederagenin and luteolin on the expression of potential targets in L02 cells. (**c**) Effects of FZHY-medicated serum on the expression of potential targets in Lx2 cells. (**d**) Effects of tanshinone IIA, hederagenin and luteolin on the expression of potential targets in LX2 cells. (**e**) Effects of FZHY-medicated serum on the collagen I and αSMA levels in Lx2 cells. (**f**) Effects of tanshinone IIA, hederagenin and luteolin on the collagen I and αSMA levels in Lx2 cells. FZHY, FZHY formula. ^#^
*p* < 0.05, ^##^
*p* < 0.01, compared with the control group. * *p* < 0.05, ** *p* < 0.01, compared with the model group. These groupings of blots are cropped from the same gel or different gels. The full-length gels and blots are included in a Appendix A. Repeated 3 times.

**Table 1 molecules-24-00338-t001:** IC50 values of major compounds for L02 and Lx2 cell lines.

Compound	IC50 (µmol/L) L02	IC50 (µmol/L) Lx2
Kaempferol	>200	196.7
Hederagenin	70.31	71.13
Luteolin	63.7	53.24
Tanshinone IIA	57.15	>200
Beta-sitosterol	>200	>200

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
