# Peer review of "Active Compounds Derived from Fuzheng Huayu Formula Protect Hepatic Parenchymal Cells from Apoptosis Based on Network Pharmacology and Transcriptomic Analysis"

_molecules, 2019, doi:10.3390/molecules24020338_

Round 1

Reviewer 1 Report

I askef for changes in the original submisdion. Thèse changes are satisfactory to me. 

Author Response

Response to Reviewer 1 Comments

Point 1: I askef for changes in the original submisdion. Thèse changes are satisfactory to me.

Response 1: Thank you.

Reviewer 2 Report

MS Title: Active compounds derived from fuzheng huayu 2 formula protect hepatic parenchymal cells from 3 apoptosis based on network pharmacology and 4 transcriptomic analysis

MS ID: molecules-424611

This an interesting study on Fuzheng huayu formula approved as antifibrotic medicine. It consists of six 60 Chinese herbs. I suggest to improve the Discussion section and try to explain which of the 60 herbs have the apoptosis activity.

Page 7. Discussion section. First paragraph.  I suggest to include the following reference:

Authors need Altamirano-Barrera A, Barranco-Fragoso B, Méndez-Sánchez N. Management strategies for liver fibrosis. Ann Hepatol. 2017;16:48-56.

Pages 7 and 8. Discussion section. The paragraphs 2-4 are very speculative. I suggest to cut or rewrite those paragraphs.

Page 10. Materials and Methods. Paragraph 4.  Authors need to indicate the Weight and age of rats used in this study

Author Response

Response to Reviewer 2 Comments

(All changes are in red)

Point 1: This an interesting study on Fuzheng huayu formula approved as antifibrotic medicine. It consists of six 60 Chinese herbs. I suggest to improve the Discussion section and try to explain which of the 60 herbs have the apoptosis activity.

Response 1: Thank you. Fuzheng huayu formula (FZHY) consists of six Chinese herbs including Salvia miltiorrhiza Bunge, fermentation mycelium powder [Cordyceps sinensis (Berk.) Sae.], Schisandra chinensis (Turcz.) Baill., pollen pini (Pinus massoniana Lamb.), semen persicae [Prunus persica (L.) Batsch] and Gynostemma pentaphyllum (Thunb.) Makino. According to our results, FZHY and its compounds such as hederagenin, luteolin and tanshinone IIA were involved in the regulation of apoptosis activity in L02 and Lx2 cells. Hederagenin is derived from semen persicae [Prunus persica (L.) Batsch]. Luteolin and tanshinone IIA are derived from Salvia miltiorrhiza Bunge. The description was added to Discussion section (Page 9, third paragraph). Thus, semen persicae [Prunus persica (L.) Batsch] and Salvia miltiorrhiza Bunge had effects on the regulation of apoptosis activity in six Chinese herbs of Fuzheng huayu formula. And the Discussion section was improved.

Point 2: Page 7. Discussion section. First paragraph.  I suggest to include the following reference: Authors need Altamirano-Barrera A, Barranco-Fragoso B, Méndez-Sánchez N. Management strategies for liver fibrosis. Ann Hepatol. 2017;16:48-56.

Response 2: Thank you. In Discussion section (First paragraph, Page 7, line 191-193), this reference and description were added.

Point 3: Pages 7 and 8. Discussion section. The paragraphs 2-4 are very speculative. I suggest to cut or rewrite those paragraphs.

Response 3: Thank you. These paragraphs were revised.

Point 4: Page 10. Materials and Methods. Paragraph 4.  Authors need to indicate the Weight and age of rats used in this study.

Response 4: Thank you. The weight and age of rats used in this study was added to Materials and Methods (Page 10, line 342).

Reviewer 3 Report

This is a comprehensive and well done and novel study demonstrating the importance of active compounds derived from fuzheng huayu formula, an ancient traditional Chinese medicine, in protecting hepatic parenchymal cells from apoptosis.The findings are partly based on network pharmacology and transcriptomic analysis. Conclusions are supported by the results. These data provided potential mechanism for protection of liver cells, in particular HPC and HSC, from apoptosis and profibrogenic activation. It also provides a potential therapeutic targets to prevent liver diseases. I have no concerns or comments. I like to congratulate the authors for conducting this novel study.

Author Response

Response to Reviewer 3 Comments

Point 1: This is a comprehensive and well done and novel study demonstrating the importance of active compounds derived from fuzheng huayu formula, an ancient traditional Chinese medicine, in protecting hepatic parenchymal cells from apoptosis.The findings are partly based on network pharmacology and transcriptomic analysis. Conclusions are supported by the results. These data provided potential mechanism for protection of liver cells, in particular HPC and HSC, from apoptosis and profibrogenic activation. It also provides a potential therapeutic targets to prevent liver diseases. I have no concerns or comments. I like to congratulate the authors for conducting this novel study.

Response 1: Thank you.

Round 2

Reviewer 2 Report

I have no comments at this time

This manuscript is a resubmission of an earlier submission. The following is a list of the peer review reports and author responses from that submission.

Round 1

Reviewer 1 Report

The aim of this study by Wu and colleagues is, as stated in the abstract, to use ‘network analysis, transcriptomic analysis, assays of cell apoptosis, viability and protein expression […] for investigating the effects and mechanisms of compounds derived from FZHY on hepatic parenchymal cell (HPC) protection and hepatic stellate cell activation.’

The manuscript is very hard to follow, it lacks information fundamental to understanding the rationale and conclusions and it cannot be read as a stand-alone paper, but needs to be read in conjunction with a previous recent paper from this group (Dong, S.; Cai, F.F.; Chen, Q.L.; Song, Y.N.; Sun, Y.; Wei, B.; Li, X.Y.; Hu, Y.Y.; Liu, P.; Su, S.B. Chinese herbal formula Fuzheng Huayu alleviates CCl4-induced liver fibrosis in rats: a transcriptomic and proteomic analysis. Acta Pharmacol Sin 2018, 39, 930-941, doi:10.1038/aps.2017.150.).

The major concern re the study is the relatively incremental advance over the previous paper from the same group, mentioned above.

Major comments:

1.       The first chapter of the results is very confusing. The dataset used is described in Dong S. et al.; 2018 doi:10.1038/aps.2017.150, but there is no information about it in this manuscript. A summary of what the dataset is should be included, e.g. what has been compared, what cells or tissues. When talking about DEGs, it would be useful to know which groups have been compared etc.

2.       With the very little information given about this dataset, it is very difficult to understand why, looking at the same data and with the same parameters (fold change > 1.5 and p-value 0.05), a different number of DEGs is obtained. It is also unclear why differential expression analysis is presented again in this paper, as it appears to be on the same dataset and with the same type of analysis and criteria as previously presented in Dong S. et al.

3.       Figure 1 also needs a lot of clarification:

-          Fig 1a: define M and F;

-          Figure 1c: define what the percentages in the Venn diagram are;

-          Figure 1d: explain what the numbers at the end of the bars represent.

4.       Re the results represented in Fig. 3, could the authors speculate on why the individual compounds inhibit viability in L02 cells, but FZHY does not?

5.       As a general comment on the manuscript, the readership of Molecules is not made only of fibrosis experts. This Reviewer feels that the introduction requires a more in depth description of the molecular mechanisms of liver fibrosis, with an explanation of what are the contributions of the different cell types (hepatic parenchymal cells and hepatic stellate cells) and chemokines (i.e. TNF-α and TGF-β). This is particularly important to then understand the experimental design, e.g. the use of Lx2 and L02 cells, and why the two different cell types are treated with different chemokines.

6.       This Reviewer could not find information on where the microarray data have been deposited.

Reviewer 2 Report

Wu et al present a paper entitled "Active compounds derived from fuzheng huayu formula protect hepatic parenchymal cells from apoptosis based on network pharmacology and transcriptomic analysis" that describes at least partly the molecular mechanisms underlying the use of a traditional chinese formula, Fuzheng huayu (FH), on hepatic parenchymal cells. They show using a combination of transcriptomics, cell biology experiments, and network analysis that FH protects parenchymal hepatic liver cells from apoptosis, and give clues on the active compounds involved in this action. The results are interesting and novel. The paper is well written, and clear.

As the journal Molecules asks authors to put "Mat and met" in the back of the paper, the results should be introduced with the model used (SD rats). here the authors start head on with the transcriptomics results and it can be confusing for the readers. 

The readers would benefit from 2 additional figures, one at the beginning of the manuscript presenting the experimental strategy and the animal model, one at the end of the manuscript summarizing the most clearly possible the sum of the results.

"mat and meth" , transcriptomics data: the authors should present more comprehensively their method (apparatus, chip etc). and normalization strategy. A previous paper is cited but as this method is really important in the paper, it should be at least summarized.

For L02 and Lx2 cells, what were the passages used for the various experiments? The authors state that experiments were repeated at least three times, I would prefer that they state in each figure the exact number of n (technical and biological)

Material and methods: please give the sex of the rats used, and justify the choice of sex.